# Intersectionality informed and narrative-shifting whole school approaches for LGBTQ+ secondary school student mental health: A UK qualitative study

Amy Morgan[1,2], Emily Cunningham[3], Juliet Dyrud[1], Liberty Elliott[1], Lauren Ige[1], Gemma Knowles[1,4], Lukasz Konieczka[5], Angela Mascolo[1], Ibrahim Sabra[1], Sara Sabra[1], E. Singh[1], Katharine A. Rimes[6], Charlotte Woodhead[1,2]*

1 ESRC Centre for Society and Mental Health, King's College London, London, United Kingdom, 2 Department of Psychological Medicine, King's College London, Institute of Psychiatry, Psychology & Neuroscience, London, United Kingdom, 3 TRIUMPH Network, MRC/CSO Social and Public Health Sciences Unit, University of Glasgow, Glasgow, United Kingdom, 4 Department of Health Services and Population Research, King's College London, Institute of Psychiatry, Psychology & Neuroscience, London, United Kingdom, 5 Mosaic LGBT+ Young Person's Trust, London, United Kingdom, 6 Department of Psychology, King's College London, Institute of Psychiatry, Psychology & Neuroscience, London, United Kingdom

* charlotte.woodhead@kcl.ac.uk

**Data Availability Statement:** There are restrictions on access to the de-identified dataset because of the sensitive nature of the topic. This includes

## Abstract

School is a key site for prevention and early intervention in public mental health, with sexual and gender minority students being a priority group for action. Context is important in understanding how school inclusion of sexual and gender minorities shapes mental health and well-being, with rapidly changing social and political forces necessitating ongoing research. This coproduced UK secondary school-based study aimed to understand (a) key components of mentally, socially and emotionally healthy school environments for LGBTQ+ students considerate of intersecting minoritised identities; (b) staff information, skills and capacity needs and (c) factors influencing uptake and implementation. Online interviews and focus groups were conducted with 63 participants (22 staff, 32 students (aged 13–19 years), and 9 training providers), diverse in relation to gender and sexual identity, ethnicity, religious and social context. Data were analysed thematically. One overarching theme captured the need for an intersectionality-informed, contextually adaptable, whole school approach which 'shifts the narrative' away from deficit thinking, challenging prevailing cis/heteronormative and White norms. This underpinned four themes: (1) 'Feeling safe, seen and celebrated: embedding intersectional signs, signals and symbols', (2) 'Everyone's business: the need for collaboration', (3) 'Embedding a culture of change', and (4) 'Re-locating the problem: challenging deficit thinking'. Contextually diverse research is needed which critically addresses ways in which social power enacted interpersonally and structurally serves to hinder schools from enacting LGBTQ+ inclusivity. Evidence to inform and develop implementation strategies for institutional changes and to advocate for wider socio-political support is also key to mitigate the potential for widening inequities linked to inequitable school environments.

details about racial and ethnic origin, and sexual orientation and gender identity of young people aged 13-18. The ethical review committee overseeing this study can be contacted at rec@kcl.ac.uk (study reference: HR-20/21-21515) for researchers who meet the criteria for access.

**Funding:** This work was supported by the TRIUMPH (Transdisciplinary Research for the Improvement of Youth Mental Public Health) Network which is funded by the Cross-Disciplinary Mental Health Network Plus initiative supported by UKRI under grant ES/S004351/1. AM, CW, and GK are part supported by the ESRC Centre for Society and Mental Health at King's College London [ES/S012567/1]. The views expressed are those of the author(s) and not necessarily those of the ESRC or King's College London. The funders had no role in study design, data collection and analysis, decision to publish, or preparation of the manuscript.

**Competing interests:** The authors have declared that no competing interests exist

## Introduction

LGBTQ+ adolescents and young people (here defined as those aged 10–24 years [1] "who identify as lesbian, gay, bisexual, transgender, and/or questioning, and/or who express diverse sexual orientations, gender identities, and/or gender expression (p.1)" [2] are a public mental health priority group [3]. Stigma, discrimination, internalised homo/bi/transphobia, bullying, and victimisation are key predictors of mental distress for LGBTQ+ young people, alongside low self-esteem, lack of acceptance and support from parents and communities [4], poorer family relationships [5] and exposure to sexual, physical, and emotional abuse [6]. In line with Minority Stress Theory [7] these experiences underscore disproportionate mental ill health, including anxiety, depression, suicidal behaviours and self-harm, substance use, and eating disorders [8].

School climate is highly influential to LGBTQ+ students' mental health, academic performance and school attendance [9–12]. LGBTQ+ young people exposed to racial/ethnic marginalisation may experience greater inequities within school environments that fail to regard their cultural contexts and intersectional identities [13]. However, research about LGBTQ+ young people from ethnically and racially minoritised groups often reinforces deficit perspectives. They perpetuate stereotypes and discrimination through an over-emphasis on mental health problems rather than normative developmental processes or positive youth development [14]. Moreover, most existing research is quantitative and US-based [15, 16] with limited UK evidence about the influence of contextual factors on effectiveness of school-based interventions to reduce mental health inequalities [17]. This reflects a broader lack of mental health interventions for sexual minority groups which considers their intersecting social identities [18]. There is also limited evidence about the experiences of students with intersecting minoritised identities [10, 19]. This is important, since research indicates the potential for school environment to constrain development of sexual and gender identity, as well as that linked to intersecting social statuses such as race/ethnicity [20].

Teachers have an important role [21], yet many remain unaware of the challenges that LGBTQ+ students face [22] and LGBTQ+ inclusion in schools is often implicitly the responsibility of LGBTQ+ staff [23]. Staff professional development about sexual and gender identity has been identified as a target to improve school climate [24, 25] but this often comprises one-off sessions [22]. Moreover, evidence is limited about how to design professional development programs [10], and about barriers to uptake and implementation [26]. Grounded in an intersectional approach and coproduced with young people, this qualitative study therefore aims to understand how to create positive school environments for LGBTQ+ students and what systemic factors influence whether, and the extent to which, schools do so.

### Theoretical framework

Intersectionality theory [27] shaped the study conceptualisation and interpretation of findings. In recognising that people occupy multiple social statuses (and that this may include a mix of advantaged and disadvantaged positions/identities), it highlights the simultaneity of intersecting forms of oppression and privilege and how these shape people's experiences (e.g., White culture, cisgenderism, hetereosexism). Queer theory [28] also informed our work, particularly at analysis and interpretation stages. Specifically, it describes three interrelated components relevant to our study, a) challenging heterosexuality as the norm ("heteronormativity"), b) challenging prevailing ideas that studies of sexual minorities should consider them as a single entity and c) emphasising the myriad ways in which race, class, culture, generation, geography and socio-political differences shapes bias against sexual minority groups. Queer theory also

challenges socially constructed binary assumptions and the oppressive power of dominant gender identity norms (cisnormativity). Drawing on these theoretical frameworks therefore frames marginality as socioculturally created rather than an inherent pathology [29]. Moreover, Moffit and colleagues [30] recognised the importance of considering how social identity categories are constructed and operationalised in research about LGBTQ+ youth, and of capturing how power shapes systems of oppression as well as privilege.

While recognising the importance of research with younger age groups, we focus on secondary/college age students (adolescents aged 13–19 years). Specifically, we address the following questions from student, staff and professional development training provider perspectives:

1.  What are key components of mentally, socially and emotionally healthy school environments for LGBTQ+ students considerate of intersecting minority identities?

2.  What are the information, skills and capacity needs of secondary school/college staff to support LGBTQ+ students' mental health?

3.  What factors influence uptake and implementation of school initiatives to help tackle inequities experienced by LGBTQ+ students?

## Methods

### Design and setting

We conducted a qualitative coproduction study as part of the Schools Training to Enhance support for LGBT+ young People Study (STEPS). Coproduction refers to "an approach in which researchers, practitioners and the public work together, sharing power and responsibility from the start to the end of the project, including the generation of knowledge."(p.5) [31]. We adopted this approach recognising young people's lived experience and expertise as central to conducting meaningful research with, rather than to, young people. STEPS aimed to explore ways to optimise professional development for staff to improve school climate to positively influence young people's mental health. All study phases (including research question/topics, planning, ethical procedures and applications, data collection, analysis and interpretation) were co-produced by a team of young people, university researchers, and LGBTQ+ organisation representatives.

### Participants and sampling

UK-based training providers of professional development training to schools/educators about LGBTQ+-related issues were initially identified through a scoping exercise. Publicly available (online) information was collected on the organisation name, type of training or resources provided and location. To maximise variation across geographic coverage, training focus, and setting (urban/rural), we invited all identified training providers.

Staff and students were purposively sampled by advertising through the coproduction teams' networks, including LGBTQ+ organisations and secondary schools (ages 13–19 years) or colleges (in the UK these specifically serve students usually between the ages of 16–19 years), and social media. Leaflets were shared digitally with potential participants stating that anyone aged 13–19 years or working in a school/college could participate. They included statements that reflected our focus on race, ethnicity, faith and rural/coastal location. The leaflets also stated that participants did not have to identify as LGBTQ+ to participate to avoid excluding those who are questioning or not 'out' who may have had concerns about taking part in an "LGBTQ+-only" study. Except for pilot focus group participants, school staff and student

participants were asked to complete an optional diversity monitoring form. Training providers were also not asked to complete the form.

## Procedure

Data were collected between 21st February-28th November 2021. Potential participants indicating interest received an e-mail invitation and information including confidentiality and withdrawal details. Up to three further contacts were made. Following informed consent, participants were assigned a unique ID number and offered £15 e-gift vouchers for their time. Participants opted to participate either via online semi-structured interview or focus group and could choose for interviews either 1-to-1 with a university researcher or 2-to-1, also with a young coproduction team member. All were audio-recorded with consent. Data were transcribed verbatim, removing identifying information.

## Topic guides

Topic guides for students, staff and training providers were developed via literature review and discussion with the coproduction team, then piloted among the coproduction team and pilot focus group with young people from an LGBTQ+ organisation (see S1 Appendix for full topic guides). The guides were adapted to improve clarity of wording, flow and length. Topics covered LGBTQ+ students' experiences at school/college (e.g., how do you think students who are, or think they might be, LGBTQ+ experience life at your school?); experiences of/with students with intersecting identities (e.g., Are there any ways that training be improved to ensure that all aspects of young people's social identities are reflected, including race and ethnicity, faith and/or socio-economic background?); mental health/well-being and support needs (e.g., What is most important in terms of helping promote positive mental health or to prevent mental health problems among LGBTQ+ pupils?); staff information, training, and capacity needs (e.g., Are there any ways that training be improved to ensure that all aspects of young people's social identities are supported?); and, take up and implementation of school initiatives (e.g., Do you think your school or staff at your school would face any challenges in taking up this type of training or putting their training into practice?).

## Data analysis

Data were inductively thematically analysed [32]. The lead author led on the analysis, with input and discussion throughout with the wider coproduction team, who familiarised themselves with a subset of the data across the three participant groups. We discussed patterns and their interpretations across three data analysis sessions, using an online collaborative tool to capture initial codes and themes. These initial insights shaped the lead authors' interpretation of the data and following further data familiarisation, remaining transcripts were descriptively coded to develop an initial coding framework, refined through iterative discussion and coding rounds. Data were imported into NVivo [33] to support data management and the coding framework applied to all transcripts. Supported by discussion and visual mapping, the team sought themes reflecting relevant key patterns within and across interviews. These were continually checked and refined against transcripts, looking for patterns and for similarities/differences within and across sample groups. Themes were defined, described and labelled, and patterns and interpretations were discussed. While we are aware of the complexities around the term 'data saturation' [34] we appraised saturation to be the point at which additional interviews, despite their richness, were no longer leading to meaningful changes to our initial coding framework. Quotes are labelled as TP (Training Provider), YP (Young Person), or SS (School Staff), self-reported sexual identity and race/ethnicity (grouped to maintain

confidentiality). Where available, we also labelled quotes with self-identified gender (labelling participants as "trans*" as an umbrella term to indicate where people have self-identified as 'man or 'woman but that this was not the same as the related 'sex' assigned at birth, alongside other self-reported gender identities, to give an indication of gender diversity within sample). We acknowledge that this may not be a term the individual may use to describe themselves.

## Reflexivity

The co-production team consisted of individuals with various perspectives based on their lived experiences from different identities, including race and ethnicity, sexual orientation, gender diversity, and age. Due to the small team size, we agreed for specific characteristics not to be disclosed to preserve anonymity. As a research team, we engaged in a continual and iterative practice of reflexivity. For instance, discussing the potential impacts of our positionality on research design, planning, data collection, analysis and writing.

## Ethical approval

Written informed consent was gathered from all participants (for those under 16 years informed assent and parental consent was obtained). Consent/assent was re-obtained verbally (audio-recorded) at the start of interviews/focus groups. Ethical approval was received from King's College London Research Ethics Committee (Palliative care, Nursing and Midwifery Research Ethics Subcommittee), reference: HR-20/21-21515.

## Results

### Sample

Qualitative data were collected from 63 participants (Table 1), including 22 school and college staff, 32 students, and 9 training providers. While we had originally intended to explore how intersections between LGBTQ+ status, race/ethnicity and religion/faith may be differentially experienced in urban and more rural areas, in practice our sample was predominantly urban, constraining our ability to explore this in depth.

We identified one overarching theme, 'A whole school, intersectional approach which shifts the narrative', and four themes: (1) 'Feeling safe, seen and celebrated: embedding intersectional signs, signals and symbols', (2) 'Everyone's business: and the importance of collaborative working', (3) 'Embedding a culture of change', and (4) 'Re-locating the problem: challenging deficit thinking'.

The overarching theme captured an underlying pattern discernible throughout the four themes: that creating mentally, socially and emotionally positive environments for LGBTQ+ students requires more fundamental and contextually contingent changes to schools' structure and culture beyond staff training alone. Specifically, data supported previous findings and practice highlighting the need for a whole school approach (WSA) -meaning action across curricula, school ethos and environment, collaboration with all staff and students, as well as family and community partnerships with schools/colleges. However, participants' accounts went further to emphasise that in WSAs, using an intersectional lens and actively shifting the narrative about LGBTQ+ (e.g., away from problematising 'LGBTQ+ issues') is necessary if such approaches are to be meaningful to young people. Specifically, by recognising that inequities remain due to White, cisgender and heteronormative structural biases inherent within educational organisations, policy and society.

**Feeling safe, seen and celebrated: Embedding intersectional signs, signals and symbols.** Many participants, across all groups, commented that the school environment and

**Table 1. Sample characteristics of students and staff for whom diversity monitoring data were collected.**

| Characteristics | | n | % |
|---|---|---|---|
| **Participant group** | Staff | 22 | 47 |
| | Students | 32 | 53 |
| **Region of residence** | North England | 2 | 4 |
| | London | 38 | 81 |
| | South England | 7 | 15 |
| **Gender identity** | Woman (including trans*) | 26 | 55 |
| | Man (including trans*) | 11 | 23 |
| | Non-binary | 7 | 15 |
| | Not sure/Don't know/Prefer not to say | 3 | 6 |
| **Gender same as assigned at birth?** | No | 11 | 23 |
| | Yes | 36 | 77 |
| **Sexual orientation** | Asexual | 3 | 6 |
| | Bisexual | 7 | 15 |
| | Gay | 5 | 11 |
| | Heterosexual or straight | 14 | 30 |
| | Lesbian | 6 | 13 |
| | Not sure/Don't know/Unsure or questioning/ Prefer not to say | 3 | 6 |
| | Pansexual | 3 | 6 |
| | Queer | 6 | 13 |
| **Ethnicity** | Asian | 10 | 21 |
| | Black | 2 | 4 |
| | Mixed/Multiple ethnic groups | 2 | 4 |
| | White British/ White other | 30 | 64 |
| | Any other group | 3 | 6 |

*Notes: Characteristics and samples have been grouped to preserve anonymity. Excludes training providers, and student pilot focus group participants.*

curriculum, often reflected cis-genderism and heteronormativity, constraining sense of belonging for those with differing experience, inhibiting self-awareness and self-acceptance. Many interviewees described attempts to counterbalance this by increasing the visibility of LGBTQ+ identities in the physical building (e.g., in LGBTQ+ inclusive posters, displays, flags, lanyards, gender neutral facilities), within other aspects of the school community/learning environment (e.g., within curricula, assemblies, societies, forums, clubs, within the staff body and external speakers), or through use of gender-neutral language, preferred names and pronouns.

> *"My old school had Pride flags along the balcony in one of the halls and it just made it seem every time you walked in there like it was a welcoming space and like the whole school reflected being a place where you felt that you were, not only like allowed to be, but celebrated for who you are. And I think those are really small things, but they can make such a huge difference in terms of how you feel."* YP25, woman, bisexual, White

> *"There are posters up throughout the school of people that anyone can talk to, including three school counsellors, on any sort of issues like that as well. So I think the school does try make it clear that there are places for people to talk about these issues and does try and to show that*

*the school is working to improve the atmosphere for people who are, who are not kind of White, cis, het, heteronormative etc."* YP22, not sure, pansexual, White

However, such experiences were not uniform and attempts at visibility commonly represented White bodies and reflected Western norms which was perceived - particularly among or on behalf of students from racialised backgrounds - as alienating, perpetuating the perception that LGBTQ+ issues did or should not relate to them, and/or limiting the relevance to their lived experience.

*"Addressing race and intersectionality through uhm, you know when you're talking about sexuality with young people is, I guess, fundamental to helping them see it themselves, I suppose, or just helping them understand themselves."* SS24, man, gay, White

"That whole thing around making intersectionality visible, uhm, making sure that if we're putting a display up about LGBT rights that we also have Black people on the display and there they are gay Muslim people, and they should know that that's acceptable.." SS23, woman, heterosexual, Asian

*"Definitely representation. I think that is so important, so making it very clear that in some communities, even though it's a minority, they still believe that being LGBTQ+ is a White phenomenon. It's just a White man's issue, it's not within other communities."* SS07, woman, queer, Black

Another key facet of visibility, particularly identified by student and training provider participants, was representation of LGBTQ+ identities and allyship within the staff body. Their accounts suggested that seeing a diversity of identity representations within the school, would enhance feelings of belonging and self-acceptance, which in turn would positively impact upon mental health and well-being. For example, by enabling young people to express and explore themselves, challenging stereotypes about 'being LGBTQ+' and what that means or looks like, facilitating young people to be open, and to seek and access support if needed. The importance of visible role-models was emphasised by the following students:

*"I don't know any, queer POC members of staff in the college. I don't know many people like with an intersecting identity (...) At least you could feel, ok, that could be me one day."* YP18, non-binary, queer, other ethnic group

*"A few of my teachers wear like queer ally badges. I think that's really nice, um 'cause obviously it shows that they're a safe person to speak to, they are a safe person to just be around, I guess. Especially since I know there are staff in school who are homophobic who, it wouldn't really be a good idea for you to talk to."* YP16, not sure, queer, White

Both staff and students recognised that the extent to which staff feel able to be openly out or in allyship may also be shaped by wider social and cultural forces, both experienced and anticipated, and by the extent to which homophobia is unchallenged within the school environment.

*"I had an incident in which - I have like the rainbow pin on my lanyard - and a student came up to me and told me, 'oh do you support these people?' and I said 'yes' and the student went, 'well, I'm homophobic and pretty much started spitting their food and when the supervisor kind of came over [they] pretty much implied that I had invited that because I should have just not had the pin, and that problem wouldn't have appeared."* SS22, woman, asexual, White

**Everyone's business: And the need for collaborative working.** In line with the 'whole school' aspect of our overarching theme, participants across all groups recognised that staff, senior leadership and governors, young people, training providers, parents and the wider community, all have a role in creating a supportive environment, while recognising that the skills, expertise and resources needed may vary by role and context.

Most staff and training providers recognised the value of external training providers to support and facilitate such collaboration. Also, the need to create spaces to share best practices, lessons learned, questions, and challenges in a way that made staff feel safe to not know (e.g., what terminology to use), make mistakes and learn. This was perceived to encourage staff to share experiences and engage more with training, as well as to gain confidence in communicating around gender and sexuality.

*"Making the training available to everybody, I think is really important. Um, and I think it's and having those conversations, um, between so that ideas can be shared in that collaborative practice. So, if somebody's feeling less confident in in responding or having those conversations, that they can seek advice and support from others and learn from each other."* SS29, woman, heterosexual, White

Participants in all groups also discussed the importance of building relationships, trust and working collaboratively with parents, carers and local communities. Building relationships (e.g., by running groups and events in local public spaces and engaging in conversations) could increase trust, awareness and understanding around sexual and gender diversity, and provide a space for transparent dialogue.

*"I think what we should be doing is reaching out to our local community and putting in place spaces where the parents can come in where the community can get involved in our conversation, because ultimately, what is a school is built by the outside community. Unless we're able to help that group of people then inside school is just gonna be kind of putting bandaids everywhere."* SS12, man, gay, White

*"I think it's really key, as well, in school to understand the wider context, so what happens outside the classroom and what happens within. Uhm, and understanding certain barriers, dynamics, uhm, what's happening in social contexts, economic contexts, specific areas, or within the college as a whole but also out in the communities. And understanding how, uhm, that might affect how people perceive and experience."* SS29, woman, heterosexual, White

In this, co-designing professional development training with schools and staff to tailor content to the local context was commonly identified by staff and training providers as key; particularly to develop skills to speak more confidently in culturally sensitive ways and to engage in 'difficult' conversations where beliefs or opinions about sexuality and gender may differ and/or within cultures where there is more stigma and less familiarity with LGBTQ+ identities.

*"Just learning about the LGBT community makes some parents really uncomfortable, so I'm not sure how the training would like go with them. It probably would be negative for them, but yeah. . .I guess a suggestion would be like maybe do like an information evening and explaining, why it's still important for their children to learn about this, these type of issues."* YP13, woman (trans*), lesbian, Black

*"I think maybe that would be quite useful, having, how to tackle difficult parents. What do you say to a parent who is so against anything LGBTQ+? Like you can't teach my child that no, they can't go to that lesson. So it might be worth that how to have these difficult conversations with parents, what do we say?"* SS08, woman, heterosexual, other ethnic group

However, it was also recognised that assuming particular parent groups/local communities were inherently less accepting or knowledgeable of LGBTQ+ identities could serve to perpetuate prejudicial views and exacerbate difference.

*"I don't want to say anything that, sounds like it's deficit ideology around what other cultures and communities might feel, so I'm not going to. Uhm, I just know that everyone's journey is different and the fact that we have less young people of colour tell us their orientation, sexual orientation or their feelings around gender, will have other factors and I'm gonna be honest with you and say that I personally don't wanna make any assumptions."* SS05, woman, heterosexual, Asian

Collaboration with students was also included within this theme, emphasising the need to ensure their perspective, experience and voices are included and valued in school initiatives, and to facilitate peer-to-peer support. Linked to the intersectional perspective, student participants highlighted the importance of engaging with a broad range of young people from different social groups.

*"Actually listening to young people, because I feel as if with adults, I do it myself at work and at school, we feel like we know the information, we feel like we've been there, don't worry, just listen to me. But listening to the young people and finding out exactly what they need and what they want."* SS07, woman, queer, Black

*"Things that work well for one, for one group of people, won't necessarily work well for another group of people. So, I think it's a case if it's speaking to like, ethnic minority, like Black and ethnic minority groups, and other faith groups and like, you know, people who live in rural areas."* YP14, non-binary, bisexual, White

**Embedding a culture of change.** This theme highlighted the importance of sustained, consistent actions which could help actively dismantle norms engrained within the school system. For example, participants across all groups talked about the importance of sustained LGBTQ+ representation–i.e., not just during Pride month; avoiding one-off training to 'tick the box'; embedding diverse LGBTQ+ identities appropriately into the curriculum across all lessons, rather than just when discussing 'LGBTQ+ issues'; and consistency of responding to homo/bi/transphobic bullying.

*"I thought that doing LGBT month will open up everybody's eyes and it'll be a situation where homophobia will disappear in a school and transphobia will disappear in school, but that's not the case. It has to be done every[day], it has to be constant drip feed."* SS07, woman, queer, Black

*"Being consistent in challenging homophobic biphobic and transphobic behaviour within the schools, that every single staff member is, you know, challenging it. It's not just that one staff member with who gets given the rainbow lanyard and you know that responsibility."* TP05

Such consistency was perceived to avoid actions being perceived as tokenistic, unauthentic and performative, and to encourage sustained iterative learning and development among staff and students to support more meaningful change. It also therefore linked to 'shifting the narrative', moving away from conceptualising intersectional 'LGBTQ+ issues' as something separate, perceived to contribute to 'othering'.

*"If you're LGBT for example, you're not just LGBT for pride month, your LGBT for the whole whole year(. . .)So, there is an initiative at the moment where we're trying to plan in points in the curriculum to, to introduce things in a normalised way rather than as a tokenistic way, I suppose."* SS28, man, gay, White

However, across groups, participant's accounts indicated that meaningful change also depends upon more fundamental shifts beyond interpersonal behaviours. For instance, proactively embedding intersectionality within initial teacher training, and making relevant school/ college values, ethos, and policies explicitly visible to staff, students and parents/communities.

*"What it isn't, is doing the blanket LGBT awareness training, ticking the box and moving on. From our perspective, it's about you know how do we actually, you know, it's about that structure of pastoral care, the ethos of the school community, your policies and your curriculum, is I think that the most effective way of actually embedding a culture of change."* TP04

Here, factors such as leadership support, school type (e.g., faith vs. secular, private vs. comprehensive) and social, economic and cultural context (e.g., rural versus inner city, local demographic profile) was seen to shape the extent to which and how schools had the capacity or will to engage with LGBTQ+ inclusion consistently.

*"I think especially since I'm in a faith school, there's just a lot of reluctancy to move with the times. It's all about being traditional, any sort of liberal step forward they take is met with resistance from, you know, the governors and parents and everything that they teach in school has to be approved by the diocese."* YP16, not sure, queer, White

*"There's those schools are rich in the terms, in terms of money and might be a state school, but demographically, they've got largely middle class, mostly White pupils, with parents who are educated, professional people, who aren't going to come charging up and they will buy it. And then the opposite end, the schools which are really poor with time and poor with resources and they've got such a lot of issues going on. So they might have quite a high BAME [Black, Asian and Minority Ethnic] proportion of pupils on roll. They are the schools that are being targeted by the government around Prevent, FGM [female genital mutilation], all of those issues are coming their way. So this [LGBTQ+] issue sits in a suite of issues that they've got to deal with, because they've got, you know, a cohort of pupils within the school that are homophobic and that might be coming from culture, but they need to do something about that."* TP01

**Re-locating the problem: Challenging deficit thinking.** Pertinent to the shifting the narrative aspect of our overarching theme, this last theme reflects a perceived need to move away from perceptions which problematise LGBTQ+ young people, to challenge deficit thinking (positioning marginalised groups as lacking and in need of 'fixing', e.g. Weiner and colleagues [35]) and instead relocate the 'problem' with cis and heteronormativity. Participants across groups discussed an over-emphasis on prejudicial talk about LGBTQ+ students. This was perpetuated in an everyday sense through slurs about gender diversity and sexual orientation

often used among students to connote negative opinions (e.g., "that's so gay"). Another example given by several students and training providers was of staff erroneously viewing a young person 'coming out' as necessarily a safeguarding issue requiring them to break confidentiality to family members, rather than only in the case in which such a disclosure would put them at or involved disclosure of risk. This was a barrier to students' accessing support but could also adversely affect students if such an involuntary 'outing' leads to familial/community rejection. Such a scenario could occur for any young person but participants particularly discussed this issue in relation to students from (or perceived to be from) cultural backgrounds less aware of, open to or supportive of sexual and gender diversity and reflected a lack of cultural sensitivity and awareness within schools.

*I used to go to quite like a very, very sort of like attempting to be secular school. Where like they would just like, try as hard they could to not sort of push like any kind of religion(...)in an attempt to be secular, they kind of like, get blinded to relevant considerations about people's religion, and I think that can put people at more of a risk of being outed to potentially unsupportive families.* YP15, non-binary, bisexual, White

*"One of the teachers actually, caused them to nearly be made homeless, because a few, more than one student was made to come out. But this particular student came from a Muslim household, and the parents were very not ok with it(....)the teacher made them come out and they were nearly kicked out their house and their church, like their mosque, yeah, because they were trans."* YP focus group participant

The following account suggests that such deficit-based assumptions may be exacerbated if staff members themselves hold beliefs that run counter to inclusive practice, and/or who are less familiar with concepts around gender and sexual identity.

*"Older teachers who, um, or staff members who, just the way that they've been brought up has been, with this with these type of issues being kind of more swept under the carpet. And so they might, just because that's been instilled in them, they might find it quite uncomfortable to have these types of conversations, and also maybe might not know kind of what's right and wrong in today's world as much as teachers that are younger know it. And maybe as well teachers from different cultures where it is less talked about too, or less socially acceptable."* YP25, woman, bisexual, White

Deficit thinking was particularly discussed in relation to gender identities due to limited staff understanding, wider societal transphobic discourse and perceptions that trans* people need to be specially accommodated (e.g., through pronouns, toilets etc). This problematises the young person and burdens them, and also reflects resistance to or lack of insight into the need to challenge underpinning norms around gender embedded within and beyond schools.

*"I know about a lot of schools that aren't progressive, who just like view sort of, like queer students, particularly like a lot of trans students I know as kind of more of a logistical problem than, like you know, treating them like a human being who deserves like, help from them. They're viewed as more of a like a sort of, you know, like puzzle, that they have to solve, in terms of their place in the school."* YP15, woman (trans*), heterosexual, White

Participants across groups commonly raised that discussing, educating, and supporting people in relation to sexual orientation and gender identity is often seen as LGBTQ+ staff and students' 'problem' and responsibility, risking exhaustion and fatigue.

*"It's particularly acute for those in the trans community, rather than the wider community because I think they've also had a burden of like feeling they need to explain to their peers as well. And like educate their peers at the same time as trying to fight, figure out who they are and who they want to be. Um, and they're kind of, the trans community within the school are also the ones who kind of are the first to have a change for uniform and the first to have a change, for toilets."* SS25, man, heterosexual, White

## Discussion

We aimed to understand student, staff and training provider perspectives on how school environments could be enhanced for LGBTQ+ students while considering intersecting minoritised identities; information, skills and capacity needs of staff; and factors influencing uptake and implementation. One overarching theme captured the need for an intersectionality-informed, contextually adaptable, whole school approach (WSA) which 'shifts the narrative' away from deficit thinking and challenges prevailing White, cis/heteronormative norms. Underpinning this were four themes: (1) 'Feeling safe, seen and celebrated: embedding intersectional signs, signals and symbols', (2) 'Everyone's business: the need for collaborative working', (3) 'Embedding a culture of change', and (4) 'Re-locating the problem: challenging deficit thinking'.

### Comparison with prior research

Our findings support the need for fundamental and proactive shifts in school culture and ethos to ensure positive environments for LGBTQ+, particularly gender diverse, students [36]. Whole School Approaches (WSA's) have been previously advocated for by practitioners in this area [37], and in recent research exploring how school climate influences LGBTQ+ student mental health [38]. Although multiple conceptualisations of WSAs exist (e.g., to anti-bulling, socio-emotional learning), our findings fit with common core principles such as the importance of students, staff, families and external stakeholders working together to embed inclusive practice within and outside of school [39]. This includes anti-bullying policy with specific regard for LGBTQ+ students [40], support spaces, as well as curriculum and staff representation of intersectional LGBTQ+ identities [37]. A meta-analysis [41] found that Whole School Approaches's to enhancing student social and emotional development showed small but significant improvements to student outcomes. Supporting our findings, Goldberg and colleagues [41] report that interventions incorporating collaborative working with the wider community including engagement with parents and local agencies were identified as more effective than those which do not. Indeed, family and peer support, access to counselling services and having trusted adults at school have been found to be protective of depression and suicide-related behaviours among LGBTQ+ students [42]. Our findings point to the importance of parental and community engagement, and to the need for WSAs to incorporate these groups, we therefore seek in further analyses to delve deeper into this specific aspect of WSAs.

Staff and students perceived lack of support, knowledge, confidence and skills to constrain communication about LGBTQ+ issues in faith schools (in the UK these are schools with formal links to specific faith-based or religious organization and/or teach the national curriculum with regard to a particular religion) and in contexts where a large proportion of the student body are from cultural backgrounds perceived to be less accepting of sexual and gender diversity. While some evidence explores this in relation to Catholic schools [43], further work is needed to unpick these concerns in greater detail to avoid generalising interpretations and perpetuating stereotypes. Nonetheless, our study supports an urgent need for LGBTQ+ school

initiatives to embed intersectional awareness and shift narratives away from deficit-perspectives which problematise LGBTQ+ experiences and identities. Not least, this is key to ensure students receive support and education which is culturally sensitive, relevant, and safe [44]. Moreover, that approaches go beyond individualised and one-off approaches involving learning about difference that perpetuate othering of sexual and gender minorities, particularly those with intersecting marginalised statuses [45]and those from faith communities [43]. Instead, approaches which critically appraise ways in which power and oppression are inherent within cis/heteronormative and White, Eurocentric systems, are needed [45].

Increasing numbers of young people self-identify as trans*, the 'debates' around gender and trans* rights are often transphobic, questioning the validity of trans* identity, and ignoring the substantial health and social inequities they face [46, 47]. Bower-Brown and colleagues [36] suggest that the "British school system is fundamentally unsuitable for non-binary and gender-questioning identities"(p.74). Our findings support this and other UK-based evidence that trans* young people may experience particular difficulties with unsupportive school climates and/or lack of staff knowledge, and that this is influenced by cis/heteronormativity and wider social discourse [48].

In line with prior research [38, 49], our findings advocate for mandatory and continuous staff professional development as a key component of any WSA. Aligned with prior research, we identified barriers to implementing /enacting LGBTQ+ affirmative initiatives such as: lack of confidence and knowledge, confusion around laws and policies and fears of parental response [50], alongside lack of government support and guidance [51]. Our study adds to this by acknowledging differences in support for, and inequities in the capacity of, schools to implementing change linked to governance (e.g., faith schools) and socio-demographic constraints. Such differences are likely to create a widening of inequities experienced by LGBTQ + students, particularly those also minoritised on the basis of religion, race, socio-economic resources, and those from Western conceptualisations of cultural contexts perceived to be less likely to be familiar or accepting of gender and sexual diversity. It is important to acknowledge that any shift away from intolerance is beneficial and that schools progress at different paces, thus change is required at multiple levels [52].

## Strengths and limitations

A strength of our study is the inclusion of multiple perspectives. While research about young people's experiences rightly should foreground their voice, it should also balance this with the need to shift responsibility from young people to those with a duty to care for them. In practice it is staff and trainers tasked with enacting change (whether passed through legislation, inter or intra-personally mediated) and thus understanding their take (while being critically aware of how their positionality shapes their viewpoints) is essential. Moreover, in coproducing our study with a team of young people we ensured that direct lived experience was foregrounded in the design, analysis and interpretation. A limitation was that we did not include the perspectives of parents/carers which, given their core role in WSAs, is an important avenue for future work. Last, while our sample overall included diversity in relation to faith, ethnicity, gender and sexual orientation, White British people comprised 56% of the staff and 60% of the student sample. It is important to critically consider this in relation to their perspectives on the experiences of students with intersecting minoritised identities and in our interpretations of this.

## Conclusion

Our findings support calls for WSAs to LGBTQ+ inclusive practice, going further to impress the need to embed intersectional experience and awareness. In doing so, we emphasise the

need to fundamentally 'shift the narrative', away from deficit thinking towards challenging complicit cis/heteronormative and White education structures. However, there is a risk of inequities being widened, particularly for LGBTQ+ young people with intersecting minoritised identities as the capacity to embed such practices are affected by school wealth, parent pressures, and socio-cultural context. Further research is needed on approaches to collaboration with a diverse range of parents and communities, and to develop evidence-based implementation strategies to support uptake and adoption of inclusive practices.

## Supporting information

**S1 Appendix. STEPS topic guides.**
(DOCX)

## Acknowledgments

We would like to thank River Újhadbor for their support with workshops and advice related to the study.

## Author Contributions

**Conceptualization:** Amy Morgan, Juliet Dyrud, Liberty Elliott, Lauren Ige, Gemma Knowles, Lukasz Konieczka, Angela Mascolo, Ibrahim Sabra, Sara Sabra, E. Singh, Katharine A. Rimes.

**Data curation:** Charlotte Woodhead.

**Formal analysis:** Amy Morgan, Emily Cunningham, Juliet Dyrud, Liberty Elliott, Lauren Ige, Angela Mascolo, Ibrahim Sabra, Sara Sabra, E. Singh, Charlotte Woodhead.

**Funding acquisition:** Juliet Dyrud, Liberty Elliott, Lauren Ige, Gemma Knowles, Lukasz Konieczka, Angela Mascolo, Ibrahim Sabra, Sara Sabra, Katharine A. Rimes, Charlotte Woodhead.

**Investigation:** Amy Morgan, Emily Cunningham, Juliet Dyrud, Liberty Elliott, Lauren Ige, Angela Mascolo, Ibrahim Sabra, Sara Sabra, E. Singh, Charlotte Woodhead.

**Methodology:** Amy Morgan, Juliet Dyrud, Liberty Elliott, Lauren Ige, Gemma Knowles, Lukasz Konieczka, Angela Mascolo, Ibrahim Sabra, Sara Sabra, E. Singh, Katharine A. Rimes, Charlotte Woodhead.

**Project administration:** Amy Morgan, Emily Cunningham, Charlotte Woodhead.

**Supervision:** Katharine A. Rimes.

**Writing – original draft:** Amy Morgan, Emily Cunningham, Charlotte Woodhead.

**Writing – review & editing:** Amy Morgan, Juliet Dyrud, Liberty Elliott, Lauren Ige, Gemma Knowles, Lukasz Konieczka, Angela Mascolo, Ibrahim Sabra, Sara Sabra, E. Singh, Katharine A. Rimes, Charlotte Woodhead.

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
