## [Decision Letter · Decision Letter 0]

10 Jan 2024

PONE-D-23-28894Intersectionality informed and narrative-shifting whole school approaches for LGBTQ+ secondary school student mental health: A UK qualitative study.PLOS ONE

Dear Dr. Woodhead,

Thank you for submitting your manuscript to PLOS ONE. After careful consideration, we feel that it has merit but does not fully meet PLOS ONE’s publication criteria as it currently stands. Therefore, we invite you to submit a revised version of the manuscript that addresses the points raised during the review process.

Thank you for this interesting submission on an important topic. In addition to addressing the comments of the reviewer (please also note the comments the reviewer made throughout the attached document in your revision), below are some additional comments for consideration:

-I believe that the introduction and theoretical framework sections could benefit from more citation and discussion of previous research involving intersectionality in LGBTQ+ youth.

-Were the authors specifically interested in particular types of intersectionality as they relate to LGBTQ+ youth? (e.g., race/ethnicity, class, rurality/urbanicity, etc)?

-More detail on the content of the interview guide would be helpful--the authors should also consider including this as an appendix in the manuscript.

We look forward to receiving your revised manuscript.

Kind regards,

Emily Lund

Academic Editor

PLOS ONE

“This work was supported by the TRIUMPH (Transdisciplinary Research for the Improvement of Youth Mental Public Health) Network which is funded by the Cross-Disciplinary Mental Health Network Plus initiative supported by UKRI under grant ES/S004351/1. AM, CW, and GK are part supported by the ESRC Centre for Society and Mental Health at King's College London [ES/S012567/1]. The views expressed are those of the author(s) and not necessarily those of the ESRC or King’s College London.”

“We would like to thank River Újhadbor for their support with workshops and advice related to the study.

This work was supported by the TRIUMPH (Transdisciplinary Research for the Improvement of Youth Mental Public Health) Network which is funded by the Cross-Disciplinary Mental Health Network Plus initiative supported by UKRI under grant ES/S004351/1. AM, CW, and GK are part supported by the ESRC Centre for Society and Mental Health at King's College London [ES/S012567/1]. The views expressed are those of the author(s) and not necessarily those of the ESRC or King’s College London.”

“This work was supported by the TRIUMPH (Transdisciplinary Research for the Improvement of Youth Mental Public Health) Network which is funded by the Cross-Disciplinary Mental Health Network Plus initiative supported by UKRI under grant ES/S004351/1. AM, CW, and GK are part supported by the ESRC Centre for Society and Mental Health at King's College London [ES/S012567/1]. The views expressed are those of the author(s) and not necessarily those of the ESRC or King’s College London.”

5. For studies involving third-party data, we encourage authors to share any data specific to their analyses that they can legally distribute. PLOS recognizes, however, that authors may be using third-party data they do not have the rights to share. When third-party data cannot be publicly shared, authors must provide all information necessary for interested researchers to apply to gain access to the data. (https://journals.plos.org/plosone/s/data-availability#loc-acceptable-data-access-restrictions)

Additional Editor Comments:

Thank you for this interesting submission on an important topic. In addition to addressing the comments of the reviewer (please also note the comments the reviewer made throughout the attached document in your revision), below are some additional comments for consideration:

-I believe that the introduction and theoretical framework sections could benefit from more citation and discussion of previous research involving intersectionality in LGBTQ+ youth.

-Were the authors specifically interested in particular types of intersectionality as they relate to LGBTQ+ youth? (e.g., race/ethnicity, class, rurality/urbanicity, etc)?

-More detail on the content of the interview guide would be helpful--the authors should also consider including this as an appendix in the manuscript.

Reviewers' comments:

Reviewer's Responses to Questions

**Comments to the Author**

1. Is the manuscript technically sound, and do the data support the conclusions?

Reviewer #1: Partly

2. Has the statistical analysis been performed appropriately and rigorously? 

Reviewer #1: N/A

3. Have the authors made all data underlying the findings in their manuscript fully available?

Reviewer #1: Yes

4. Is the manuscript presented in an intelligible fashion and written in standard English?

Reviewer #1: No

5. Review Comments to the Author

Reviewer #1: 1. Language and grammar corrections should be made, as well as edited for clarity. This paper does not seem well-edited for spelling, grammar, and syntax with several errors throughout. There are several sentence fragments and run-ons.

2. Vague declarations can be improved through specification and supporting evidence. Several topics and terms are brought up with no reference to definitions or reasons for methodological choices.

3. There was confusion regarding who exactly was sampled, their ages, and whether or not they were LGBTQ+.

4. Methodological discussion was sound, with need for rationale/discussion of chosen theory to support choices.

5. Quotes included important themes not necessarily discussed by authors. Context and discussions of how many participants discussed each topic could be helpful.

6. Overall, a lack of definitions and consistency is creating for a confusing narrative which, at this point, does not present clear goals and results that could impact the community.

This manuscript has good potential, but gets lost in the grammar mistakes and lack of clarity. See comments attached.

6. PLOS authors have the option to publish the peer review history of their article (what does this mean?). If published, this will include your full peer review and any attached files.

Reviewer #1: **Yes: **Dannie Klooster, MS

---

## [Author Response · Author response to Decision Letter 0]

30 Jan 2024

Dear reviewers,

Thank you for your considered, helpful and clear comments on our paper entitled, “Intersectionality informed and narrative-shifting whole school approaches for LGBTQ+ secondary school student mental health: A UK qualitative study”. We have made tracked changes in the revised document in response to the comments on grammar and clarification of terms (e.g., ‘young people’), as well as on content. We focus on the content comments here for clarity.

1. More information about coproduction as an approach. 

Thank you, we agree and have added a definition of coproduction, as well as brief rationale for taking this approach in research with young people (p.5 lines 17-21)

2. Add %/numbers of how many people expressed which topics for reference. Thank you for this suggestion which we do understand. However, we have chosen not to report percentages as it suggests a more positivist approach than that taken. This may have been appropriate if we had used an approach such as content analysis but we consider it incompatible with the reflexive thematic analytic approach we took. For example, see for a discussion:

Neale, J., Miller, P., & West, R. (2014). Reporting quantitative information in qualitative research: guidance for authors and reviewers. Addiction, 109, 175–176. Available: https://onlinelibrary.wiley.com/doi/pdf/10.1111/add.12408

3. Page 14, line 2. “could add more context here about cultural differences in school settings, parent-teacher/staff interactions, and WSAs' approach to these discussions.” We very much agree that these points merit more detailed and nuanced discussion. For the purposes of this paper and the need to balance succinctness in word count with detail, we decided to take an approach which provided more of an overarching perspective. We aim to offer a second paper which delves more deeply into context, parent/teacher interactions and WSA approaches to community and parental engagement. To acknowledge this, we have added a line into the discussion (p.21 lines 10-12)

4. On terminology (female’/male vs woman/man) we agree, thank you. This initially reflected how the diversity monitoring form was structured but agree that it is inappropriate, so have amended to ‘woman’ and ‘man’ throughout.

5. Clarify ‘deficit thinking’. Thank you we have added a reference and description (p.18 lines 1-2)

In addition, we note and appreciate the additional comments provided by the Editor. Please find below our responses to these:

6. I believe that the introduction and theoretical framework sections could benefit from more citation and discussion of previous research involving intersectionality in LGBTQ+ youth.

Thank you, we were mindful of the length of submission and of providing a background that would speak to research about school experiences in particular. We have gladly added the following references which together provide a broader perspective, while acknowledging the limitations in scope to offer a comprehensive review of the literature.

Moffitt U, Juang LP, Syed M. Intersectionality and youth identity development research in Europe. Frontiers in psychology. 2020 Jan 31;11:78.

Huang YT, Ma YT, Craig SL, Wong DF, Forth MW. How intersectional are mental health interventions for sexual minority people? A systematic review. LGBT health. 2020 Jul 1;7(5):220-36.

Kim SE, Toomey RB, Anhalt K. Latinx sexual minority youth's identity development and experiences with preparation for bias. Family Relations. 2023 72(3): 948-65. 

7. Were the authors specifically interested in particular types of intersectionality as they relate to LGBTQ+ youth? (e.g., race/ethnicity, class, rurality/urbanicity, etc)?

Thank you for this clarification. Our study predominantly focused on intersections between race/ethnicity (and/or) religion/faith and LGBTQ+ status. While we originally intended to explore how these intersections may be differentially experienced in urban and more rural areas, in practice our sample was predominantly urban. We have added clarification in the results (p.8 line 24).

8. More detail on the content of the interview guide would be helpful--the authors should also consider including this as an appendix in the manuscript.

Thank you, we agree this would provide important detail and context to the results. We therefore have added further detail in the methods section including example questions from each topic guide (p.7 lines 7-20), as well as submitted the topic guide as an appendix.

---

## [Decision Letter · Decision Letter 1]

22 Mar 2024

PONE-D-23-28894R1Intersectionality informed and narrative-shifting whole school approaches for LGBTQ+ secondary school student mental health: A UK qualitative study.PLOS ONE

Dear Dr. Woodhead,

Thank you for submitting your manuscript to PLOS ONE. After careful consideration, we feel that it has merit but does not fully meet PLOS ONE’s publication criteria as it currently stands. Therefore, we invite you to submit a revised version of the manuscript that addresses the points raised during the review process.

The authors are advised to conduct a thorough revision of the manuscript to improve readability

We look forward to receiving your revised manuscript.

Kind regards,

Ricardo de Mattos Russo Rafael, Ph.D.

Academic Editor

PLOS ONE

Journal Requirements:

Reviewers' comments:

Reviewer's Responses to Questions

**Comments to the Author**

1. If the authors have adequately addressed your comments raised in a previous round of review and you feel that this manuscript is now acceptable for publication, you may indicate that here to bypass the “Comments to the Author” section, enter your conflict of interest statement in the “Confidential to Editor” section, and submit your "Accept" recommendation.

Reviewer #1: All comments have been addressed

2. Is the manuscript technically sound, and do the data support the conclusions?

Reviewer #1: Yes

3. Has the statistical analysis been performed appropriately and rigorously? 

Reviewer #1: N/A

4. Have the authors made all data underlying the findings in their manuscript fully available?

Reviewer #1: Yes

5. Is the manuscript presented in an intelligible fashion and written in standard English?

Reviewer #1: Yes

6. Review Comments to the Author

Reviewer #1: The authors have done an excellent job addressing the majority of the issues brought up and created a stronger, more organized manuscript. The minor revisions in the document attached zero in on grammar errors (missing comments, run-on sentences, missing words/phrases). After these minor changes are made, the document should be ready for publication.

Excellent work!

7. PLOS authors have the option to publish the peer review history of their article (what does this mean?). If published, this will include your full peer review and any attached files.

Reviewer #1: **Yes: **Dannie Klooster

---

## [Author Response · Author response to Decision Letter 1]

13 May 2024

Thank you for your feedback on the revisions made to our paper, ‘Intersectionality informed and narrative-shifting whole school approaches for LGBTQ+ secondary school student mental health: A UK qualitative study.” (Reference PONE-D-23-28894R1). Please find below a point-by-point response to the issues raised in your e-mail.

The authors are advised to conduct a thorough revision of the manuscript to improve readability

Thank you, we have re-read the manuscript to appraise readability and note that we have responded to and revised all suggestions for wording, grammar and phrasing made by the reviewer. We are confident that the manuscript is readable as it stands.

Please review your reference list to ensure that it is complete and correct. If you have cited papers that have been retracted, please include the rationale for doing so in the manuscript text, or remove these references and replace them with relevant current references. Any changes to the reference list should be mentioned in the rebuttal letter that accompanies your revised manuscript. If you need to cite a retracted article, indicate the article’s retracted status in the References list and also include a citation and full reference for the retraction notice

Thank you, we have checked our references and there are no retracted references. We have made several revisions to the referencing format after checking against the PLOS ONE reference guide to ensure they are all complete and correct.

Reviewer #1: The authors have done an excellent job addressing the majority of the issues brought up and created a stronger, more organized manuscript. The minor revisions in the document attached zero in on grammar errors (missing comments, run-on sentences, missing words/phrases). After these minor changes are made, the document should be ready for publication.

Excellent work!

Thank you for your helpful feedback on the manuscript and revisions – we are grateful for your comments and have made those changes suggested in the revised manuscript.

---

## [Decision Letter · Decision Letter 2]

25 Jun 2024

Intersectionality informed and narrative-shifting whole school approaches for LGBTQ+ secondary school student mental health: A UK qualitative study.

PONE-D-23-28894R2

Dear Dr. Woodhead,

We’re pleased to inform you that your manuscript has been judged scientifically suitable for publication and will be formally accepted for publication once it meets all outstanding technical requirements.

Kind regards,

Ricardo de Mattos Russo Rafael, Ph.D.

Academic Editor

PLOS ONE

Reviewers' comments:

Reviewer's Responses to Questions

**Comments to the Author**

1. If the authors have adequately addressed your comments raised in a previous round of review and you feel that this manuscript is now acceptable for publication, you may indicate that here to bypass the “Comments to the Author” section, enter your conflict of interest statement in the “Confidential to Editor” section, and submit your "Accept" recommendation.

Reviewer #1: All comments have been addressed

Reviewer #2: All comments have been addressed

2. Is the manuscript technically sound, and do the data support the conclusions?

Reviewer #1: Yes

Reviewer #2: Yes

3. Has the statistical analysis been performed appropriately and rigorously? 

Reviewer #1: N/A

Reviewer #2: N/A

4. Have the authors made all data underlying the findings in their manuscript fully available?

Reviewer #1: Yes

Reviewer #2: Yes

5. Is the manuscript presented in an intelligible fashion and written in standard English?

Reviewer #1: Yes

Reviewer #2: Yes

6. Review Comments to the Author

Reviewer #1: Thank you for addressing all comments and your patience in waiting for feedback. Excellent work, congratulations!

Reviewer #2: (No Response)

7. PLOS authors have the option to publish the peer review history of their article (what does this mean?). If published, this will include your full peer review and any attached files.

Reviewer #1: **Yes: **Dannie Klooster

Reviewer #2: No

---

## [Editor Report · Acceptance letter]

2 Jul 2024

PONE-D-23-28894R2 

PLOS ONE

Dear Dr. Woodhead, 

I'm pleased to inform you that your manuscript has been deemed suitable for publication in PLOS ONE. Congratulations! Your manuscript is now being handed over to our production team.

Kind regards, 

on behalf of

Dr. Ricardo de Mattos Russo Rafael 

Academic Editor

PLOS ONE